# Design of an Air Suction Wheel-Hole Single Seed Drill for a Wheat Plot Dibbler

Xinchun Ma [1,2], Qixiang Gong [1,2], Qingjie Wang [1,2,*], Dijuan Xu [3], Yinggang Zhou [4], Guibin Chen [1,2], Xinpeng Cao [1,2] and Longbao Wang [1,2]

1 College of Engineering, China Agricultural University, Beijing 100083, China
2 Key Laboratory of Agricultural Equipment for Conservation Tillage, Ministry of Agricultural and Rural Affairs, Beijing 100083, China
3 College of Mechanical and Electrical Engineering, Beijing Vocational College of Agriculture, Beijing 100083, China
4 Institute of Urban Agriculture, Chinese Academy of Agricultural Sciences, Chengdu 610299, China
* Correspondence: wangqingjie@cau.edu.cn

**Abstract:** Focusing on the problems of the poor filling ability and stability of the mechanical wheat seeder and the complicated structure of the pneumatic seeder, a special air suction wheel-hole single seed drill for remote controlled self-propelled single seed dibbler in wheat plots was designed in this paper. According to the agronomic requirements of precision seeding in wheat plots, the seeding wheel radius was set at 180 mm 16 suction holes. Using the discrete element simulation software EDEM to analyze the seed disturbance effect of different parameter designs, the thickness of seed suction ring was 16 mm, the height of seed suction mouth was 4.5 mm, and the diameter of seed suction cam was 12 mm. Through hydrodynamic simulation, the phase angle of the negative pressure chamber was 280 degrees, positive pressure chamber was 22 degrees, phase angle of the unpressurized interval zone was 20 degrees, thickness of the negative pressure chamber was 24.5 mm, diameter of transition pipe was 17.5 mm and length of the transition pipe was 14.5 mm. Based on the above design parameters, the samples were then processed and benchtop experiments carried out. The results showed that under the best operating parameters, the re-suction index was 0.82%, the leakage index was 6.67%, and the qualified index was 92.41%, which met the design requirements. This study could provide a reference for the design of single-seed dibbling technology in wheat plots.

**Keywords:** wheat; air suction wheel-hole; seed drill; discrete element simulation; single-seed dibbling

## 1. Introduction

Wheat plot breeding experiments are an important way of selecting good varieties, plays an important role in increasing wheat yield and improving wheat quality. Single seed dibbling can reduce the influence of objective factors such as nutrient and light competition on the differences in various varieties, ensure the consistency of wheat growth status, and better select superior varieties [1–3]. In China, most plot breeding experiments are still conducted by adopting artificial seeding to realize single-seed dibbling [4,5], and the consistency of experiment conditions is poor, which seriously affects the efficiency and quality of plot breeding experiments [6]. In the process of mechanized single seed dibbling, seeds are evenly discharged from a seed drill and distributed in a plot of specified length with a certain row spacing, which can improve the accuracy of the seeding link of the plot breeding experiment and ensure the scientific authenticity of the results [7–11]. As a core component of a seeding operation, the seed drill plays a vital role in improving seeding quality. Wheat plot breeding experiments requires a high uniformity of sowing depth and grain spacing [12–15]. At present, the lack of a special single seed drill is an important factor in restricting the mechanization of single seed dibblers in wheat plots [16].

At present, the research on single seed dibbling technologies is still focused on crops with larger grains, such as corn and soybean, while crops with small irregular seeds, such as wheat, are less studied with research into wheat single seed drills still in the experimental stage [17–20]. The Austrian Wintersteiger company [21] developed a combined wheat precision seed drill with a radial groove seed metering disk and a spiral groove seed metering disk for wheat plot breeding experiments; however, its structure is complex and operation velocity low. Zhang et al. [22,23] designed a wheat nest hole-wheel precision drill with a "cylindrical body and spherical base" wheel-hole, which ensured the uniformity of seed spacing of the seed drill, and partly improved the uneven seeding of the precision seed drill in a wheat plot. Liu et al. [24] designed a ring-belt precision seed drill in a wheat plot, designed the hole parameters of the hole belt, and selected the appropriate hole belt materials, which provided a unique design idea for precision seeding in wheat plots. Liu et al. [25] designed a pneumatic pinhole wheel type wheat precision seeding device and proposed a wheat precision seeding method based on single grain separation and low-level stable seeding, suitable for single grain precision seeders in wheat plots. Cheng et al. [26] designed an air suction combined wheat precision seed drill which adopted the working principle of airflow-hole combination. The combination of airflow negative pressure seed suction and hole-filling can make a drill improve single seed filling performance.

The above theoretical research and related experiments provided research ideas for the development of the wheat single seed dibbling technology. The seed filling performance of mechanical wheat seed drills is poor, which easily causes wheat grain damage, poor seed metering stability, and difficult to achieve single seed dibbling. Though pneumatic seed drills can protect wheat seeds well and keep seed filling and metering performances stable, most of them are very complex [27]. Currently, there are still no mature pneumatic wheat single seed drills.

This work designs an air suction wheel-hole single seed drill to solve these above problems. It realizes the single seed dibbling of wheat in plot breeding experiments well. Through theoretical analysis and simulation experiments, the structural parameters of the seed drill were determined. The optimal working parameters of the seed drill were determined through benchtop experiments, and the actual working performance was verified through field experiments. The results may provide a reference for the future study of single seed dibbling technologies in wheat plots.

## 2. Materials and Methods

### 2.1. Overall Structure and Working Principle

#### 2.1.1. Integral Structure of the Air Suction Wheel-Hole Single Seed Drill

Precision seeding in wheat plots requires strict uniformity of single grain rate and grain spacing of single seeding [28]. To design such a machine it is necessary to ensure its seeding stability and avoid uneven seeding caused by changes in the working environment. In addition, the seeds are required to be evenly arranged and in a fully constrained state before being discharged, and the seed injury rate is low in the process of seed taking and seed throwing to reduce seed jumping during seed dropping. Therefore, combined with the existing wheat air suction seed drill, this paper designed an air suction wheel-hole single seed drill. Its structure is shown in Figure 1.

The seed drill was composed of a seed-metering wheel and a pneumatic distributor wheel. Two air chambers were completely separated by sealing rings to ensure the stable distributor of air force without leakage. There was a support shaft in the middle of the seed metering wheel, which generated relative rolling between the two bearings and the pneumatic distributor wheel so that the seed metering holes appeared alternately in the negative pressure and the positive pressure air chambers in turn. The cover of the seed drill isolated the internal structure of the seed drill from the outside, preventing soil from entering and damaging the precision structure during field work. The bearing pedestal and the seed metering shaft had three functions. Firstly, the seed drill was connected with the lifting support of the single seed dibbler through the bearing pedestal, so the

pneumatic distributor wheel was fixed. Secondly, the bearing pedestal cooperated with the seed metering shaft to limit the axial position of the seed metering wheel and pneumatic distributor wheel, to ensure the stability of the air chamber distribution. Thirdly, inputting power through the seed metering shaft to drive the seed metering wheel to rotate.

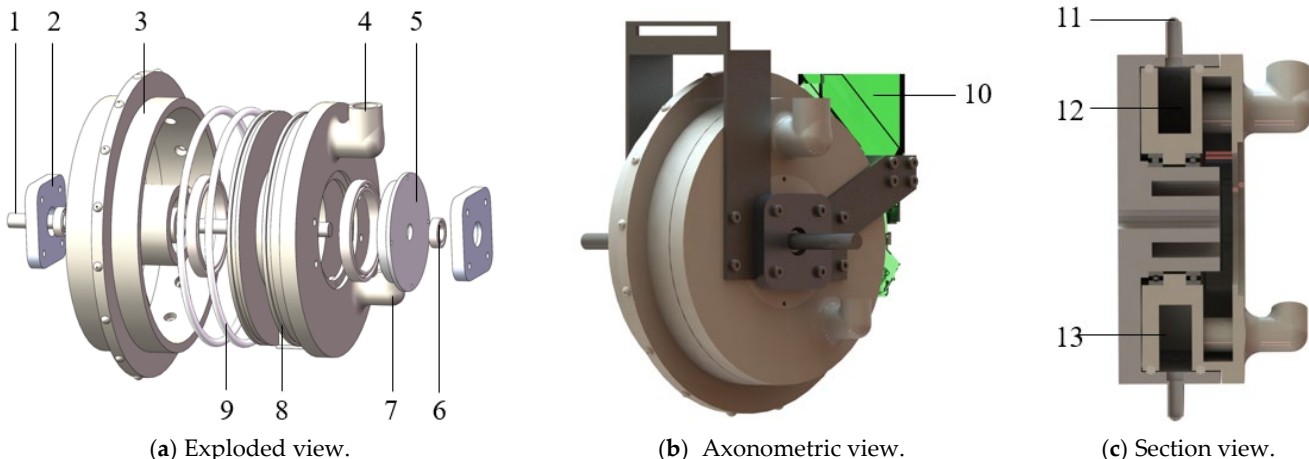

(**a**) Exploded view.　　(**b**) Axonometric view.　　(**c**) Section view.

**Figure 1.** Structure of the air suction wheel-hole single seed drill. Note: 1, Seed metering shaft; 2, Bearing pedestal; 3, Seed metering wheel; 4, Air suction port; 5, Seed metering cover; 6, Bearing; 7, Air blowing port; 8, Pneumatic distributor wheel; 9, Sealing ring; 10, Seed box; 11, Seed suction hole; 12, Negative pressure chamber; 13, Positive pressure chamber.

2.1.2. Working Principle

As shown in Figure 2, the seed metering operation of the air suction wheel-hole single seed drill can be divided into four processes, including seed suction, seed clearing, seed carrying and seed feeding. As shown in Figure 3, the pneumatic distributor wheel was fixed on the lifting frame and the inner part of the seed drill divided into the negative pressure and positive pressure air chambers. Driven by the seed metering shaft, the seed metering wheel rotates at a uniform velocity, and several seed suction holes are evenly distributed at the periphery of the seed metering wheel. When the seed metering wheel rotates to the seed box area, it enters the seed suction area I, and the seed suction hole stably absorbs seeds under the action of the negative pressure pneumatic force. The wheel then rotates through the seed suction area and enters the seed cleaning area II. The absorbed wheat seeds are impacted by the positive air flow beam. If a seed suction hole contains more than one seed, the absorption force of non-dominant seeds is less than the impact force, and the excess seeds are washed back into the seed box to ensure no more than one seed is in the seed suction hole, thus realizes the separation of single seeds. The seed metering hole can continuously obtain stable negative pressure conditions from the negative pressure air chamber in the seed carrying area III, stably adsorb the seeds to the bottom of the seed metering wheel, and when it reaches the designated seed feeding area IV, the negative pressure air chamber is isolated, the seed suction hole enters the sealed area, the seeds lose negative pressure, and are put into the seed trench under the action of gravity. Due to the seed suction hole being below ground level and close to the bottom of the seed trench, the seeds do not bounce after landing. The seed metering wheel continues to rotate and enters the air blowing area V after passing through the sealing area. Under the action of positive pressure air flow, the air suction hole prevents soil from entering the air suction port, cleans the area near the air suction hole, and completes a seeding process.

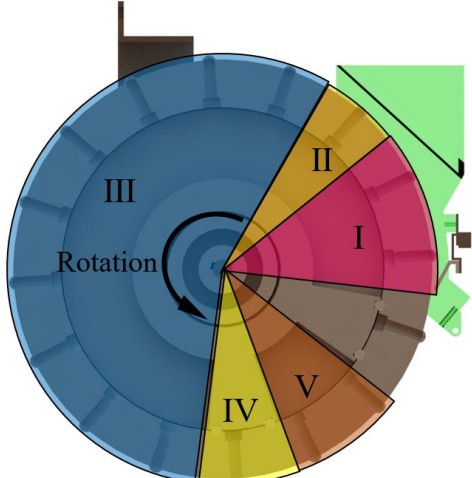

**Figure 2.** Seeding process. I. Seed suction area; II. Seed cleaning area; III. Seed carrying area; IV. Seed feeding area; V. Air blowing area.

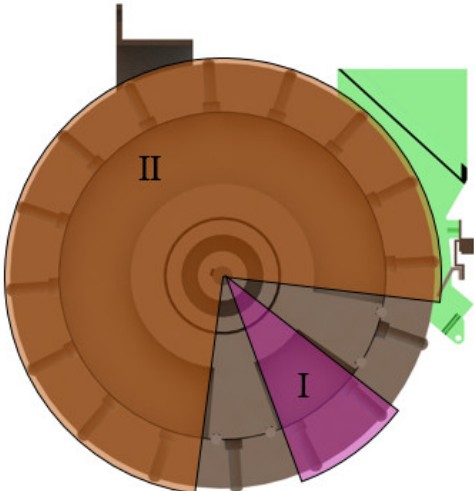

**Figure 3.** Air chamber distribution. I. Positive pressure air chamber; II. Negative pressure air chamber.

*2.2. Design of Relevant Parameters of the Seed Metering Wheel*

The seed metering wheel is the main executive part of the seed drill, its structure is shown in Figure 4. The central shaft of seed metering wheel was adopted with a hollow structure to minimize weight to ensure sufficient strength. The inner hole of the central shaft was connected with the seed metering axle through a flat key to ensure power input and the outer ring of the shaft was connected with the pneumatic distribution wheel through two bearings to ensure coaxial rotation and axial limitation. To reduce the height difference between the seed feeding position and the bottom of the seed ditch, the air suction convex ring structure was designed to ensure that the seed feeding position was close to the bottom of seed ditch. The limit wheel shoulder was designed to not only play a limiting role in the sowing process, but also complete the preliminary soil covering after sowing.

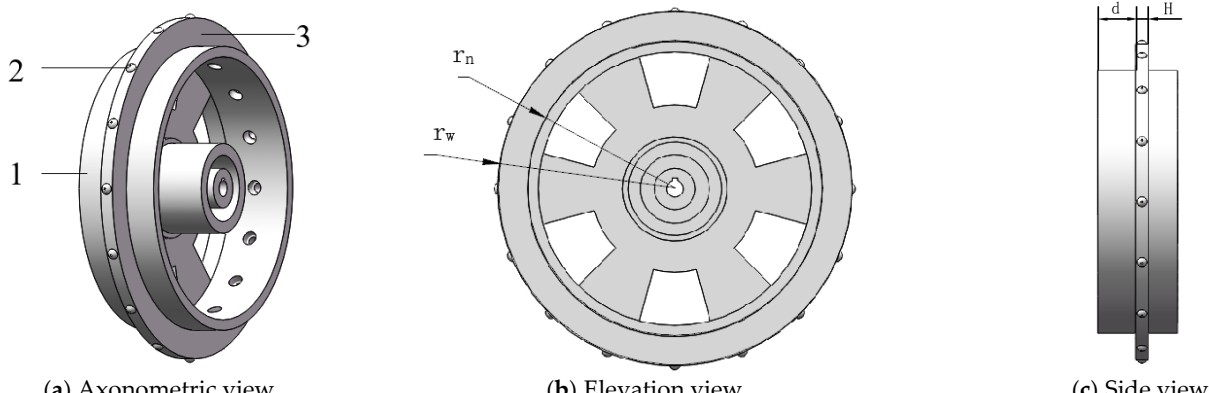

| (**a**) Axonometric view. | (**b**) Elevation view. | (**c**) Side view. |

**Figure 4.** Seeding process. Note: 1, Limiting wheel shoulder; 2, Suction port bulge; 3, Suction convex ring.

2.2.1. Design of the Number of Seed Suction Holes

The number of seed suction holes is closely related to the seed spacing, the diameter of seed drill, the rotation velocity of the seed metering wheel and the forward velocity of machine. In order to avoid sliding and bouncing caused by the initial velocity of the seed in seed feeding process, it is necessary to ensure that the relative ground velocity when the seed leaves the seed drill is zero. Thus, the linear velocity at the seed metering hole must be equal to the forward velocity of single seed dibbler. At this time point, the arc length between two adjacent seed suction holes is equal to the seed spacing of sowing, thus:

$$N = \frac{2\pi r_w}{l_o} \tag{1}$$

$$\alpha = \frac{l_o}{r_w} \tag{2}$$

$$l_n = l_{n1} + (v_{ve} - v_{me})\frac{\alpha}{\omega} \tag{3}$$

where $N$ is the number of seed suction holes; $r_w$ is the radius of seed metering wheel, cm; $l_0$ is the designed grain spacing, cm; $l_n$ is the actual seed spacing, cm; $l_{n1}$ is the actual design grain spacing, cm; $\alpha$ is the phase difference between two adjacent suction holes, rad; $\omega$ is the angular velocity of the seed metering wheel, rad/s; $v_{me}$ is the forward velocity of the single grain dibbler, m/s; $v_{ve}$ is the speed of the seed metering wheel, m/s.

The grain spacing was taken as 3.0 ~ 7.5 cm. It can be seen from Equation (3) that in order to obtain a larger seed spacing, one can only increase the forward velocity of the machine or reduce the rotational velocity of the seed metering wheel. To obtain smaller grain spacing, one must do the opposite operation. Compared with increasing the rotational velocity of the seed metering wheel, increasing the velocity of the machine consumes more energy, the balance of the machine becomes worse, and the forward velocity of the single seed dibbler can reach the upper limit threshold. Therefore, choosing the maximum seed spacing as the design basis, and the seed spacing $l_o$ as 7.0 cm. Then, in the design of $r_w$, is thus:

$$h_t = r_w - r_n \tag{4}$$

where $h_t$ is the height of the air suction convex ring, mm; and $r_n$ is the radius of the limiting wheel shoulder, mm.

The designed sowing depth was 0~5.0 cm. To prevent the seed suction hole from being blocked by soil, ensuring the seed suction hole was close to the bottom of seed ditch, sufficient allowance was reserved. Additionally, the height difference $h_t$ between the air suction convex ring and the limiting wheel shoulder was taken as 30 mm. Considering the overall size of the single seed dibbler and the working parameters of the fan, the size of the seed metering wheel should not be too large. On the premise of ensuring sufficient

installation space, the radius of the limiting wheel shoulder $r_n$ was 150 mm, and the radius of the seed metering wheel $r_w$ was 180 mm. Then, the number of air suction holes of the seed metering wheel was rounded to 16, and the actual design grain spacing was:

$$l_{n1} = \frac{200\pi r_w}{N} \tag{5}$$

After calculation, $l_{n1}$ was 7.06. At this time, $\alpha$ was 22.5°, and the actual design particle spacing was rounded to 7 cm, which met the design requirements.

### 2.2.2. Parameter Design of Seed Suction Bump

The seed suction bump is the bearing platform of the seed suction hole, which provides a platform for the adsorbed seeds in the seed carrying stage. Additionally, in the seed feeding stage, it can deliver wheat seeds to a height closer to the bottom of seed ditch. Meanwhile, it also plays a role in disturbing the seeds in the seed filling stage. In addition to the seed drill material and the rotational velocity of seed metering wheel, the main parameters affecting the seed disturbing effect include the thickness of the seed suction convex ring $H$, the convex height of the seed suction mouth $h$, and the convex diameter of the seed suction mouth $d$.

The operational velocity of the single seed dibbler was 1.2~2.6 km/h, and the rotational velocity of the seed metering wheel was calculated as 1.67~4.01 rad/s under the conditions of zero-velocity seeding. In the actual operational process, due to the influence of ground conditions and other factors, the forward velocity of the machine would be affected, so the actual range of seed metering velocities was larger than this range. Considering material weight, installation size and seed ditch size of the seed drill, the thickness of the seed suction bump should be as small as possible. However, the friction force of the seed suction bump on the wheat seeds was the main driving force to disturb the seeds, in order to ensure the stability of seed adsorption, the minimum thickness of the seed suction convex ring should be larger than 2 times the length of long axis of wheat, thus the minimum was 12 mm. According to the size of the ditcher, the maximum thickness was 16 mm. The bulge of the seed suction mouth had an upward disturbing force on wheat seeds, and the greater its height, the more obvious the effect was. However, if it was more than 2 times the short axis length of wheat seeds, it would store seeds easily, so the maximum was 6 mm, and the minimum was 1.5 times the short axis length, that was 3 mm. If the diameter of the bulge of seed suction port was larger than the thickness of seed suction bump, a gentle slope would be formed near seed suction port to reduce the disturbance of the bulge to the seed. Additionally, if the bulge was too small, it would not work either. Its effect was related to the thickness of the seed suction convex ring, so was set as follows:

$$d = k_i H \tag{6}$$

where $k_i$ stands the relationship coefficient between the thickness of the seed suction convex ring and the convex diameter of the seed suction mouth, and was 0.55~0.75.

### 2.2.3. Determination of Wheat Seed Parameters

The physical characteristics of wheat seeds directly affects the seed suction effect of the seed metering wheel. Its triaxial size not only affects the main parameters of the seed metering wheel, but also provides model parameters and boundary conditions for simulation experiments [29]. The experiment used *Jimai 22* wheat seed as the object, and the relevant parameters were determined. Wheat seeds were randomly selected, damaged and small seeds were removed, and 150 seeds were selected from them and their three axial dimensions were measured with vernier calipers, as shown in Table 1. If the longest dimension was the length $L$ of the long axis, the shortest dimension was the thickness $H$ of

the short axis, and the middle dimension was the width *B* of the middle axis, its equivalent diameter was:

$$D_z = \sqrt[3]{LBH} \tag{7}$$

**Table 1.** Determination results of the triaxial size of wheat seeds.

| Parameters | Maximum | Minimum | Standard Deviation | Average Value |
|---|---|---|---|---|
| Length | 7.42 | 5.64 | 0.63 | 6.05 |
| Width | 4.10 | 2.53 | 0.47 | 3.32 |
| Thickness | 3.82 | 2.86 | 0.35 | 3.15 |
| Equivalent diameter | 4.43 | 3.44 | 0.36 | 3.88 |

The results show that the average long axis, middle axis and short axis of *Jimai 22* were 6.05 mm, 3.32 mm and 3.15 mm, respectively.

### 2.3. Design of Relevant Parameters of the Seed Metering Wheel

The pneumatic distributor wheel is the main undertaker of the air chamber distribution of seed drill, but the whole air chamber is formed by the air suction path of seed metering wheel and the air chamber of the pneumatic distribution wheel. The whole air chamber could be divided into three parts: negative pressure chamber, positive pressure chamber and non-pressure area. The negative pressure chamber provided continuous and stable negative pressure for the seed suction hole, covered the seed suction area, seed cleaning area and seed carrying area, and was responsible for the stable adsorption of seeds. The positive pressure chamber was located at the back of the seed feeding area, and the coverage area was small, mainly to prevent the blockage of the seed suction hole. To separate negative pressure chamber and positive pressure chamber, a sealed non-pressure area was designed to prevent cross-air.

#### 2.3.1. Overall Structural Design of the Pneumatic Distribution Wheel

The distribution of the air chamber was mainly realized by the pneumatic distributor wheel. Its structure is shown in Figure 5. The pneumatic distribution wheel adopted evacuation printing. The bearing limiting shoulder axially limited the two bearings connected with the seed metering wheel to ensure the stability of the seed metering operation. The air suction port and air blowing port were designed in an upward direction, convenient for connection and installation with the fan.

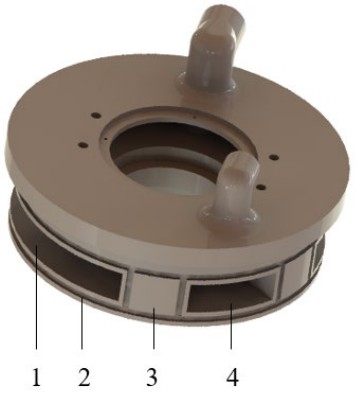

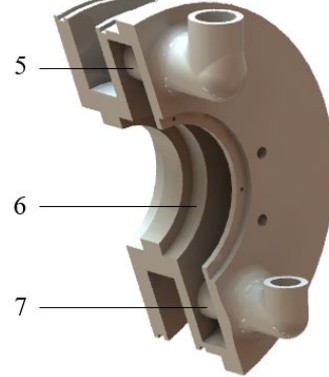

(**a**) Oblique side view of the pneumatic distributor wheel.    (**b**) Sectional view of the pneumatic distributor wheel.

**Figure 5.** Structural diagram of the pneumatic distributor wheel. Note: 1, Negative pressure air chamber; 2, Seal ring groove; 3, Non-pressure sealing area; 4, Positive pressure chamber; 5, Air suction channel; 6, Bearing limiting shoulder; 7, Air blowing channel.

Two non-pressure sealing areas divided the whole air chamber into two parts: negative pressure and positive air pressure chambers, as shown in Figures 6 and 7.

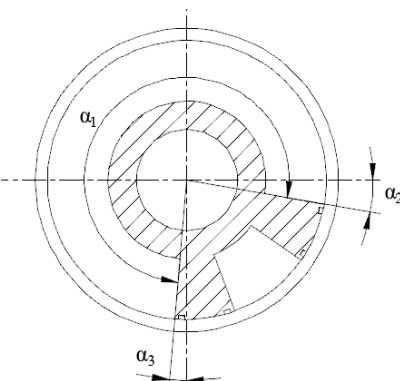

**Figure 6.** Distribution of negative pressure air chamber.

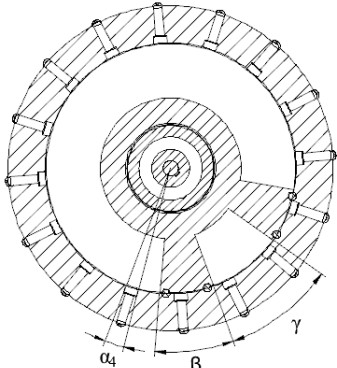

**Figure 7.** Distribution of positive pressure chamber.

The phase angle of the negative pressure chamber $\alpha_1$, included the angle between the negative pressure starting position line and the ending position line, covering an area larger than the seed suction area, the seed carrying area and the seed feeding area. $\alpha_2$ was the angle between the starting position of the negative pressure chamber and the horizontal line, that was, the seed suction advance angle. $\alpha_3$ was the angle between the end position of the negative pressure air chamber and the plumb line. To ensure the seed drill had sufficient seed suction time, the air suction port should maintain a stable negative pressure before entering the seed box and set the advance angle $\alpha_2$ to 15°. $\alpha_3$ affects the seed feeding position. In this paper, it was designed to be 5° [9], thus:

$$\alpha_1 = 270° + \alpha_2 - \alpha_3 \tag{8}$$

At this time, the phase angle of the negative pressure chamber $\alpha_1$ was 280 °.

The process for the air suction hole to pass through the interval between positive pressure chamber and negative pressure chamber was asymptotic. Once the round mouth of the air suction hole entered this area, a connected air path was formed. If two air holes were in contact at the same time within one interval, the positive and negative pressure chambers would connect, causing cross-air, affecting the tightness of air chamber during operation and the seed suction effect. Therefore, the phase angle of the air chamber spacing angle $\beta$ should not be too high, and its minimum value should be enough to completely cover one air suction hole, thus:

$$\beta_{\max} \leq \alpha \tag{9}$$

$$\alpha_4 \leq \beta_{mix} \tag{10}$$

$$\alpha_4 = \frac{180° d_x}{\pi r_f} \tag{11}$$

where $\alpha$ is the spacing angle of the seed suction hole, (°); $\alpha_4$ is the phase angle corresponding to the circular surface of the seed suction hole at the graduation, (°); $d_x$ is the diameter of the seed suction hole, mm; $r_f$ is the radius of the graduation circle of the seed suction hole, mm.

In this paper, the diameter of seed suction hole was 15 mm, and the radius of graduation circle was 138 mm, thus $6.23° \le \beta \le 22.5°$. Considering the stability of the chamber and the allowance of the sealing ring, $\beta$ 20° was selected.

The positive pressure air chamber should ensure that all air suction holes in the soil layer maintain positive pressure air flow to avoid being blocked by the soil, as shown in Figure 8, and thus:

$$\gamma = \arccos\theta\left(\frac{r_n}{r_w}\right) - \alpha_4 \tag{12}$$

The phase angle of the positive pressure cavity was calculated to be 22°.

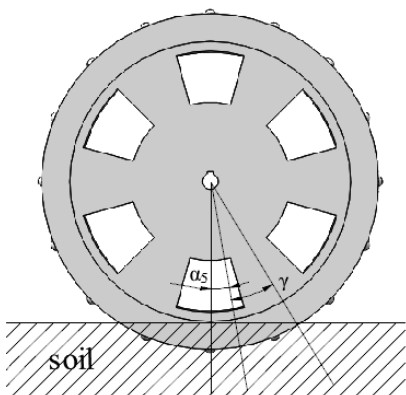

**Figure 8.** Schematic diagram of the seed metering wheel in soil.

### 2.3.2. Parameter Design of the Air Chamber of the Seed Drill

To design the parameters of the air chamber of the seed drill, the negative pressure about the seed suction and seed carrying processes must be specified first. In the seed suction process of the seed drill, wheat seeds were interfered with by the seed group, while the external force of the wheat seeds was reduced in the seed carrying process, so the adsorption force required by the seeds was smaller. Therefore, the negative pressure value was at its maximum during the seed suction process. The seed force during seed suction is shown in Figure 9 and Equation (13):

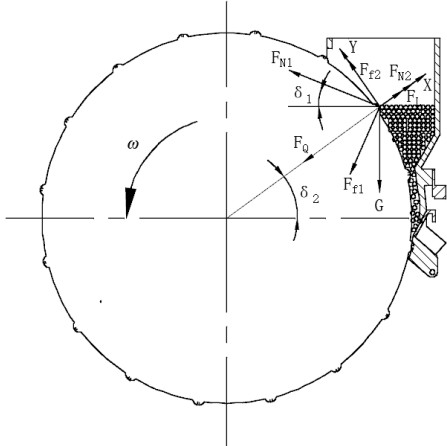

**Figure 9.** Force analysis of seed during seed suction.

$$\begin{cases} F_Q + F_{f1}\sin(\delta_1 + \delta_2) + F_{N1}\cos(\delta_1 + \delta_2) + G\sin\delta_2 = F_L + F_{N2} \\ F_{f2} + F_{N1}\sin(\delta_1 + \delta_2) = G\cos\delta_2 + F_{f1}\cos(\delta_1 + \delta_2) \\ F_{f1} = F_{N1}f_1 \\ F_{f2} = F_{N2}f_2 \\ F_L = m\omega^2 r_w \end{cases} \tag{13}$$

where $F_Q$ is the adsorption force of the negative pressure air flow of the seed, N; $G$ is the gravity of the seed, N; $F_L$ is the centripetal force required for the seed to make a uniform circular motion, N; $F_{f1}$ is the frictional force between the seed group and the seed, N; $F_{N1}$ is the support of the seed groups to the seeds, N; $F_{f2}$ is the frictional force between the seed suction port and the seed, N; $F_{N2}$ is the support of the seed suction port to the seeds, N; $f_1$ is the dynamic friction coefficient between the seed group and the seed; $f_2$ is the dynamic friction coefficient between the seed suction port and the seed; $\delta_1$ is the angle (°) between seed support and horizontal direction; $\delta_2$ is the angle (°) between the adsorption force on the seed and the horizontal direction.

According to Equation (13):

$$F_Q = \frac{\pi d_1^2 P_X}{4} = m\omega^2 r_w + F_{N2} - F_{f1}\sin(\delta_1 + \delta_2) - F_{N1}\cos(\delta_1 + \delta_2) - G\sin\delta_2 \tag{14}$$

where $d_1$ is the diameter of the seed suction port, m; and $P_X$ is the negative pressure value, kPa.

The actual working effect of the seed drill in the field was affected by the environmental conditions, and the seed drill had a complex air path structure. When the negative pressure air flow inputted from the air suction port reached the seed suction port, there would be a certain pneumatic loss [9], thus:

$$P_X \geq \frac{4K_w\left[m\omega^2 r_w + F_{N2} - F_{f1}\sin(\delta_1 + \delta_2) - F_{N1}\cos(\delta_1 + \delta_2) - G\sin\delta_2\right]}{\pi d_1^2} \tag{15}$$

$$P \geq K_b P_X \tag{16}$$

where $K_w$ ($K_w$ = 2.2~3.4) is the working stability coefficient of the seed drill, which is affected by the working environment; $P$ is the input negative pressure value of the air suction port, kPa; and $Kb$ ($Kb$ = 4.8~6.2) is the pneumatic compensation coefficient. To ensure sufficient negative pressure for seed suction, the minimum value of $P$ was taken as −5 kPa.

The air flow stability of the seed drill was mainly affected by the parameters of the negative pressure air chamber and air suction pipe. The structure of the negative pressure air chamber is shown in Figure 10. The larger the diameter of the air suction hole, the greater the adsorption force it could provide. To prevent wheat seeds from entering the air suction pipe, the maximum diameter of seed suction hole was 2 mm. The seed suction pipe was affected by the thickness of convex ring inside the seed suction convex ring. To ensure sufficient material strength, the diameter $d_2$ of the seed suction pipe was 10 mm. To reduce the loss of air force in the pipe and ensure the stability of the negative pressure, a transition pipe was added between the air chamber and seed suction pipe. To obtain a larger air inlet, the diameter $D$ should be less than the thickness of the air chamber, and a certain margin should be reserved. The length $l$ should be as large as possible under the strength of material. The larger the thickness of the air chamber and volume of the chamber, the more stable the air pressure was, but the greater the power pressure of the fan was required. Therefore, a value of 20~25 mm was chosen, thus the value range of the transition pipe diameter $D$ was 14~18 mm, and the value range of the length $l$ was 10~15 mm.

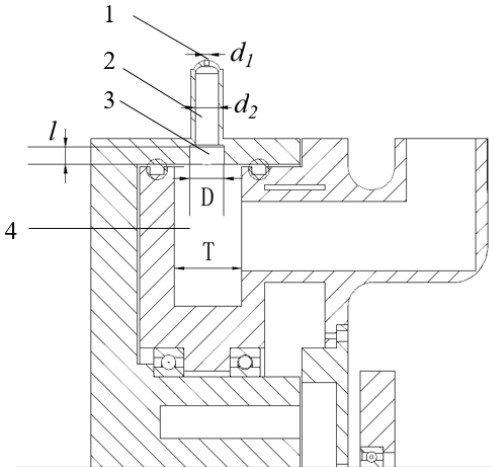

**Figure 10.** Negative pressure air chamber structure. Note: 1, Seed suction hole; 2, Seed suction pipe; 3, Transition pipe; 4, Negative pressure air chamber.

## 3. Results and Analysis

### 3.1. Simulation Analysis of the Seed Drill

3.1.1. Selection of the Simulation Analysis Parameters

Using the EDEM discrete element simulation software to simulate and analyze the disturbance of wheat seeds in the seed metering wheel, the average velocity of the seed group was used as the basis for evaluating the seed disturbance effect of the seed metering wheel [30]. A simplified model of wheat seeds was established, as shown in Figures 11 and 12.

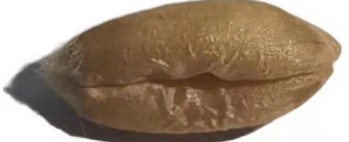

**Figure 11.** Physical picture of a wheat seed.

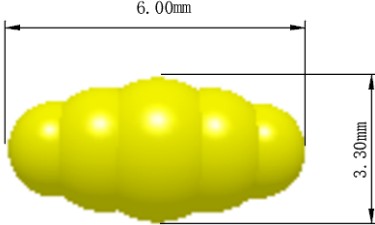

**Figure 12.** Simplified model of a wheat seed.

The length of model was 6.00 mm, and the maximum diameter was 3.30 mm. This simulation only observed the disturbance of seeds of the seed metering wheel, it did not consider the damage to the seeds, but only established a seed-particle model. The seed metering wheel and seed box were made of resin, and the specific simulation parameters are shown in Table 2 [31,32].

**Table 2.** Main parameters of discrete element simulation.

| Factors | Parameters | Value |
|---|---|---|
| Wheat seed properties | Poisson's ratio | 0.35 |
| | Young's modulus/MPa | 275 |
| | Density/(g/cm$^3$) | 1.68 |
| Seed metering wheel attribute | Poisson's ratio | 0.44 |
| | Young's modulus/MPa | 2590 |
| | Density/(g/cm$^3$) | 1.16 |
| Seed box properties | Poisson's ratio | 0.25 |
| | Young's modulus/MPa | 2170 |
| | Density/(g/cm$^3$) | 2.85 |
| Wheat seeds-wheat seeds | Impact recovery coefficient | 0.168 |
| | Static friction coefficient | 0.652 |
| | Dynamic friction coefficient | 0.076 |
| Wheat seeds– seed metering wheel | Impact recovery coefficient | 0.424 |
| | Static friction coefficient | 0.586 |
| | Dynamic friction coefficient | 0.072 |
| Wheat seeds-seed box | Impact recovery coefficient | 0.710 |
| | Static friction coefficient | 0.614 |
| | Dynamic friction coefficient | 0.070 |

3.1.2. Simulation Results

In order to reduce the times of the experiments, three-factor and three-level orthogonal experiments were designed. The experiment factors and codes are shown in Table 3, and the experiment scheme design and results are shown in Table 4.

**Table 3.** Experiment factors and level codes.

| Level Codes | Factors | | |
|---|---|---|---|
| | Thickness of Seed Suction Convex Ring/$H$ (mm) | Bulge Height of Seed Suction Mouth/$h$ (mm) | Coefficient/$k_i$ |
| 1 | 12 | 3 | 0.55 |
| 2 | 14 | 4.5 | 0.65 |
| 3 | 16 | 6 | 0.75 |

**Table 4.** Experiment scheme design and results.

| Experiment No. | Experiment Factor and Level | | | Average Velocity of Seed Group (m/s) |
|---|---|---|---|---|
| | $H$ | $h$ | $k_i$ | |
| 1 | 1 | 1 | 1 | 0.0764 |
| 2 | 1 | 2 | 3 | 0.0848 |
| 3 | 1 | 3 | 2 | 0.0791 |
| 4 | 2 | 1 | 3 | 0.0952 |
| 5 | 2 | 2 | 2 | 0.0848 |
| 6 | 2 | 3 | 1 | 0.0860 |
| 7 | 3 | 1 | 2 | 0.0892 |
| 8 | 3 | 2 | 1 | 0.0942 |
| 9 | 3 | 3 | 3 | 0.0935 |

The effect of the 4th group of the simulation experiment is shown in Figure 13. Different colors represent different seed velocities. The lighter the color, the faster the seed velocity. It can be seen from Figure 13a that the seed disturbance effect of the seed metering wheel was obvious. The seeds in the seed box moved along the seed metering axle for at

least one interval phase angle under the action of the seed suction convex ring and seed suction mouth bulge. The seed group disturbance was sufficient, which could provide good seed suction conditions. As shown in Figure 13b, the disturbance of the seed suction convex ring to the seed group mainly depended on the friction of the plane, which could only disturb the seeds which were in contact with the ring. In Figure 13c, the disturbance of seeds by bulge could disturb the deeper seeds, and the effect was more obvious. However, if the size was not appropriate it was easy for the situation shown in Figure 13d to occur; a groove-like structure was formed between the seed suction mouth bulge and the ring surface of the suction bulge, which may cause some wheat seeds to stay here and affect the seed cleaning effect.

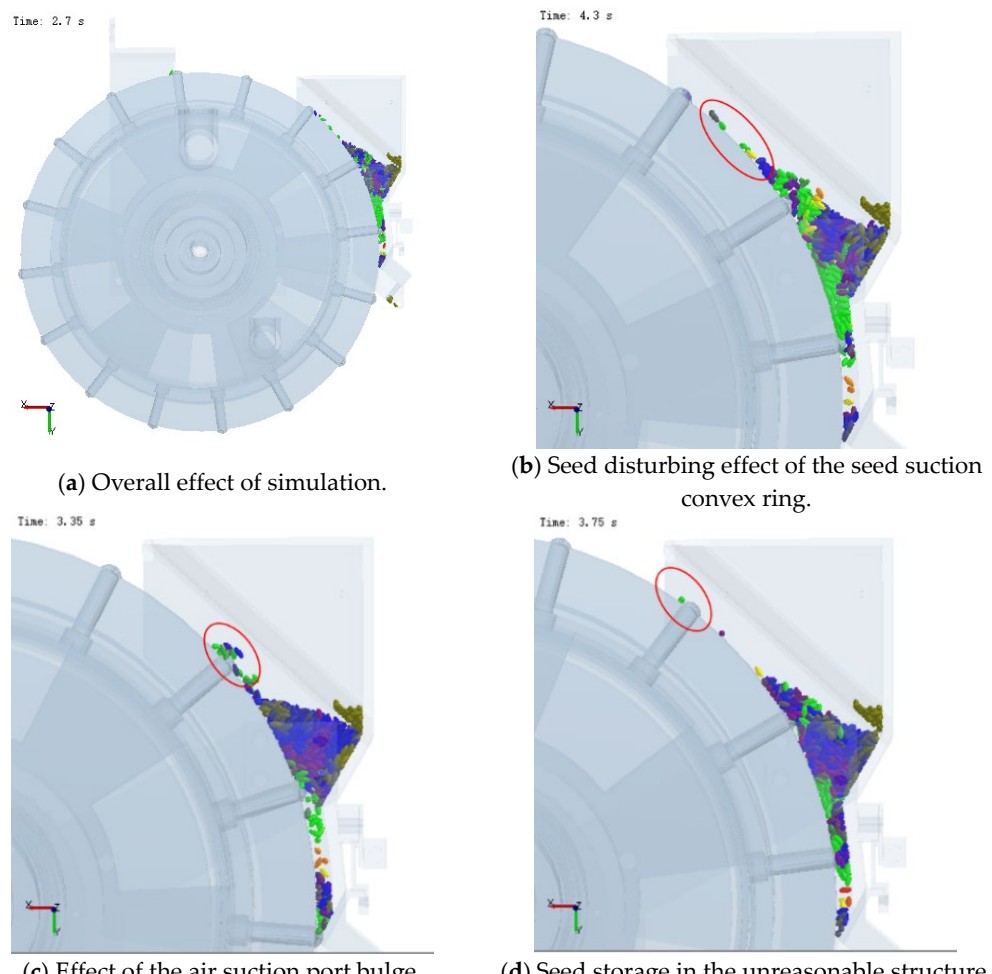

(**a**) Overall effect of simulation.

(**b**) Seed disturbing effect of the seed suction convex ring.

(**c**) Effect of the air suction port bulge.

(**d**) Seed storage in the unreasonable structure.

**Figure 13.** Discrete element simulation process.

The simulation process was designed for 10 s, and the step interval was 0.01 s, so the velocity of all seeds at each time point could be obtained. The average velocity of all seeds at each time point was taken as the instantaneous velocity of the seed group at that time. The data analysis software was used to obtain the instantaneous velocity of each experiment seed group at every 0.01 s in 10 s, and then obtained velocity of the seed group in 10 s. The line chart of experiment results of each group is shown in Figure 14.

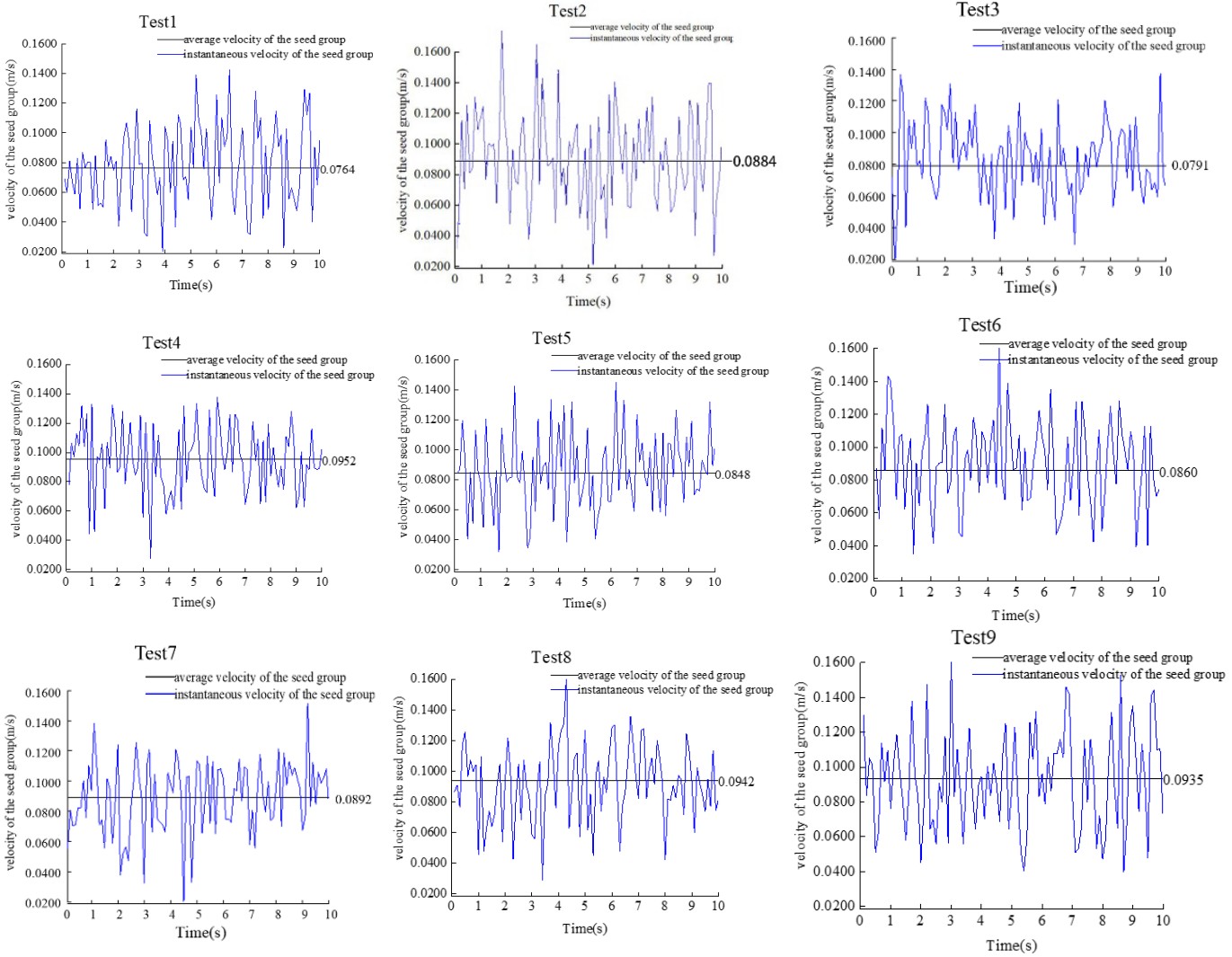

**Figure 14.** Line chart of experiment's data of each group.

### 3.1.3. Analysis of the Simulation Data

The results were analyzed by range, as shown in Table 5.

**Table 5.** The range analysis table.

| Analysis Factors | Average Velocity of Seed Group (m/s) | | |
|:---:|:---:|:---:|:---:|
| | $H$ | $h$ | $k_i$ |
| $k_1$ | 0.0801 | 0.0869 | 0.0855 |
| $k_2$ | 0.0887 | 0.0879 | 0.0844 |
| $k_3$ | 0.0923 | 0.0862 | 0.0912 |
| The optimal array | $H_3\ h_2\ k_{i3}$ | | |

The greater the average velocity of the seed group, the better the seed disturbance effect of the outer wheel of the seed drill. At the same time, according to the line chart of the results, the distribution of the seed velocities of the seed group could be observed. The more concentrated the velocity distribution, the more stable the seed disturbance effect was. According to the results and considering the overall size of the design parameters, we chose $H_3\ H_2\ K_{i3}$ arrays. The thickness $H$ of the seed suction convex ring was 16 mm, the height $h$ of the seed suction mouth was 4.5 mm, and the correlation coefficient $k_i$ was 0.75. Additionally, the diameter of the seed suction cam was 12 mm.

*3.2. Simulation Analysis of the Air Chamber*

3.2.1. Selection of the Simulation Analysis Parameters

In order to further explore the influence of the parameters of the negative pressure air chamber and air suction pipe on the seed suction effect, the ANSYS Work Bench simulation platform was used to conduct a hydrodynamics simulation, analyze the air flow in the negative pressure chamber, and select the optimal parameter array. Experiment factors and levels codes are shown in Table 6:

**Table 6.** Experiment factors and level codes.

| | Factors | | |
| --- | --- | --- | --- |
| **Level Codes** | **Thickness of Negative Pressure Chamber/*T* (mm)** | **Diameter of Transition Pipe /*D* (mm)** | **Length of Transition Pipe /*l* (mm)** |
| 1 | 22 | 14 | 11 |
| 2 | 24 | 16 | 13 |
| 3 | 26 | 18 | 15 |

The extract and mesh the fluid domain of the seed metering drill through the ANSYA Mesh platform is shown in Figures 15 and 16. The fluid domain was divided into about 6 million meshes, with an average horizontal to vertical ratio of 3, an opposite side deviation angle of 0.5, and a high-quality mesh accounting for 96.58%. The quality of mesh dividing was good. The inlet boundary layer mesh of the suction pipe was divided, and the mesh structure was optimized to achieve higher quality simulation results.

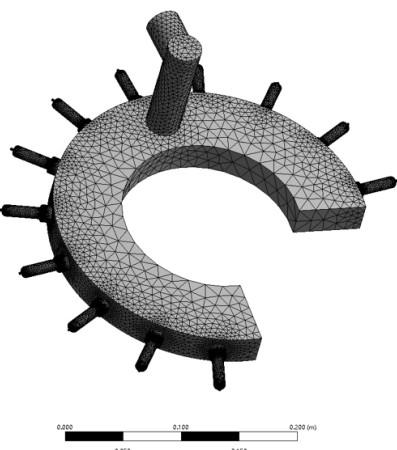

**Figure 15.** Mesh dividing in the fluid domain.

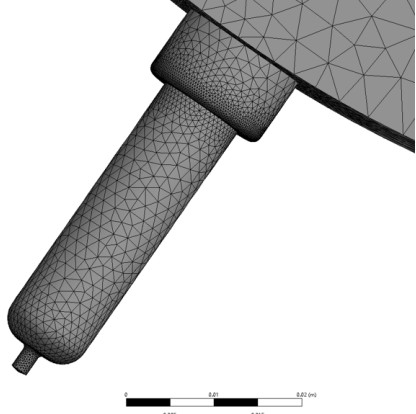

**Figure 16.** Mesh dividing of the boundary layer of seed suction pipe.

The divided fluid domain was imported into the ANSYS Fluent module, and the standard K-epsilon turbulence model was selected. Subsequently, the air pressure boundary condition was adopted, the inlet pressure was −5 kpa, the turbulence intensity was 5%, the air flow diameter was 2 mm, and the outlet boundary pressure was free. As shown in Figure 17, the interface was built at the center of the seed drill suction hole as the monitoring object, and the air flow rate at the seed suction hole was selected as the evaluation basis of the simulation experiment. After the simulation calculations converged, the average flow velocity of the air flow from the seed suction hole obtained from the post-processing software was be used as the experiment result.

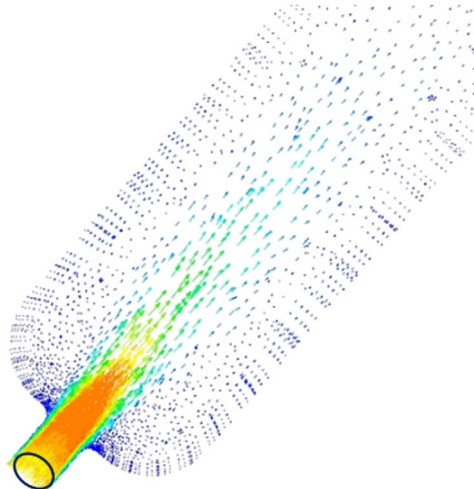

**Figure 17.** Monitoring of air flow rate at the seed suction hole.

### 3.2.2. The Simulation Results

We designed a three-factor and three-level quadratic orthogonal rotation experiment scheme and took four significant digits, the simulation results are shown in Table 7.

**Table 7.** Simulation experiment scheme and results.

| Experiment No. | Experiment Factors and Levels | | | Average Air Velocity of Seed Suction Hole/$Y$ (m/s) |
| --- | --- | --- | --- | --- |
| | $T$ | $D$ | $l$ | |
| 1 | 3 | 2 | 3 | 71.32 |
| 2 | 2 | 2 | 2 | 70.02 |
| 3 | 3 | 3 | 2 | 70.96 |
| 4 | 1 | 1 | 2 | 67.94 |
| 5 | 1 | 2 | 3 | 68.75 |
| 6 | 3 | 1 | 2 | 69.84 |
| 7 | 2 | 2 | 2 | 69.30 |
| 8 | 3 | 2 | 1 | 70.26 |
| 9 | 2 | 3 | 1 | 69.35 |
| 10 | 2 | 2 | 2 | 69.24 |
| 11 | 2 | 2 | 2 | 69.18 |
| 12 | 2 | 3 | 3 | 71.56 |
| 13 | 2 | 1 | 1 | 68.98 |
| 14 | 1 | 3 | 2 | 69.14 |
| 15 | 1 | 2 | 1 | 68.52 |
| 16 | 2 | 2 | 2 | 70.02 |
| 17 | 2 | 1 | 3 | 68.83 |

After the simulation, we took the 16th group experiment as an example, used the post-processing software ANSYS results to obtain an airflow velocity cloud chart and line chart, as shown in Figures 18 and 19. In the process of seed metering, there were

12 seed suction pipes in the negative pressure area at the same time. During the simulation experiment, the air flow velocity of each seed suction pipe was obtained, equivalent to the working state of each seed suction pipe at different phase angles of the negative pressure chamber in the working process. It can be seen from the flow velocity cloud that the air force was concentrated at the seed suction port, the flow velocity was much larger than the surrounding space, and the air force concentration point was small, which could effectively use the negative pressure provided by the negative pressure chamber. The flow rates of the different seed suction pipes, i.e., seed suction pipes in different phases, were almost the same, which indicated that the overall air force distribution of the negative pressure chamber was uniform.

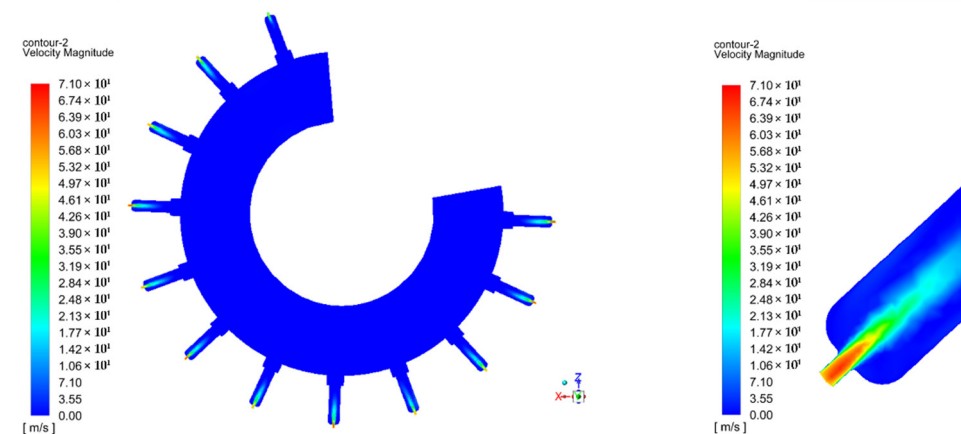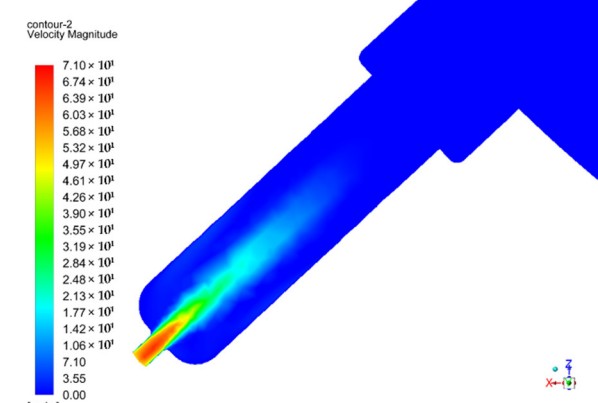

(**a**) Cloud chart of airflow velocity in the negative pressure chamber.

(**b**) Cloud chart of airflow velocity in a seed suction hole.

**Figure 18.** Cloud chart of airflow velocity.

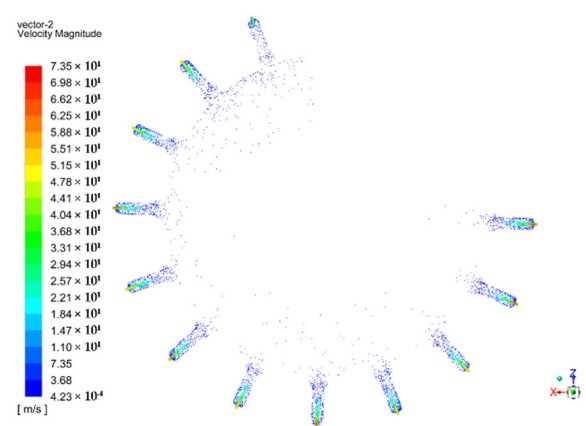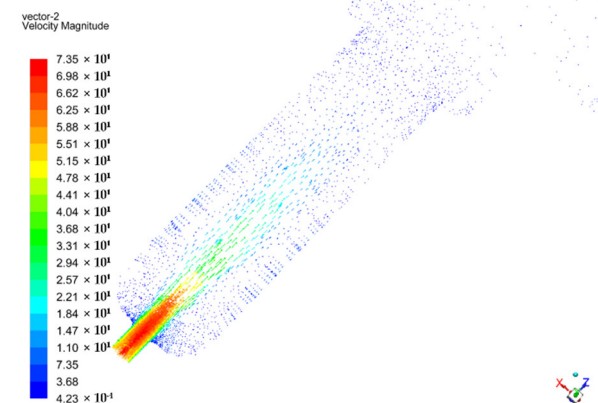

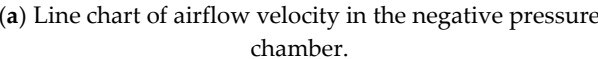

(**a**) Line chart of airflow velocity in the negative pressure chamber.

(**b**) Line chart of airflow velocity in a seed suction hole.

**Figure 19.** Line chart of airflow velocity.

The post-processing software was used to analyze the flow traces of air flow at the air inlet and seed suction port, the flow traces of the negative pressure chamber and seed suction pipe were obtained as shown in Figure 20. After entering the negative pressure chamber from the air inlet, the flow was evenly distributed along the inner wall of the negative pressure chamber. The air was divided near the position where the air suction pipe contacted the negative pressure chamber and entered the air suction pipe through the transition pipe. The air flow in the seed suction pipe was stable without vortex, and the transition pipe had no obvious effect on receiving and transmitting the air flow. The

overall air flow was stable. In negative pressure chamber there were less vortexes at places far from the air inlet, and air pressure losses. Therefore, the air inlet was arranged near the seed suction area. Furthermore, some vortexes were generated at the edge of the negative pressure chamber wall, especially at the connection of the chamber wall. Thus, fillets should be adopted into the design to reduce vortexes caused by the sudden turning of the air flow and improve the pressure stability of the negative pressure chamber.

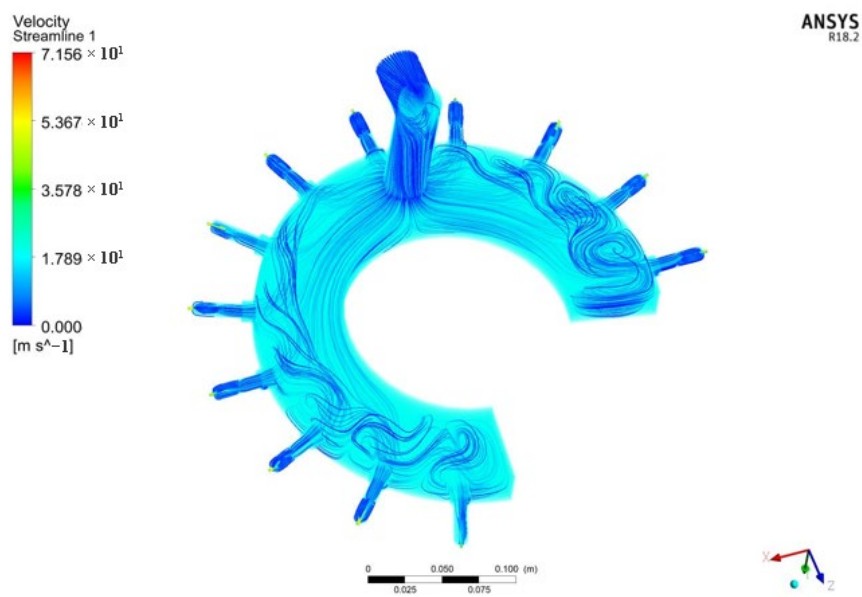

**Figure 20.** Flow trace chart of the fluid domain.

### 3.2.3. Analysis of the Simulation Data

Using the Design-Expert data analysis software, the simulation results were subjected to quadratic regression analysis, and the variance analysis results are shown in Table 8.

**Table 8.** Variance analysis of the airflow velocity.

| Source of Variation | Sum of Squares | Mean Square | *F* Values | *p* Values |
| --- | --- | --- | --- | --- |
| Models | 14.856/14.532 | 1.650/3.632 | 12.412/34.593 | 0.0016 ***/<0.0001 *** |
| $T$ | 8.060/8.060 | 8.060/8.060 | 60.605/76.764 | 0.0001 ***/<0.0001 *** |
| $D$ | 3.672/3.672 | 3.672/3.672 | 27.61034.975 | 0.0012 ***/<0.0001 *** |
| $l$ | 1.402/1.402 | 1.402/1.402 | 10.547/13.362 | 0.0141 **/0.0033 *** |
| $T_D$ | 0.001 | 0.001 | 0.012 | 0.9157 ns |
| $T_l$ | 0.172 | 0.172 | 1.294 | 0.2926 ns |
| $D_l$ | 1.392/1.392 | 1.392/1.392 | 10.469/13.264 | 0.0143 **/0.0034 *** |
| $T_2$ | 0.002 | 0.002 | 0.019 | 0.8932 ns |
| $D_2$ | 0.013 | 0.013 | 0.103 | 0.7568 ns |
| $l_2$ | 0.144 | 0.144 | 1.086 | 0.3319 ns |
| Residual | 0.930/1.260 | 0.132/0.113 | | |
| Lack of fit | 0.193/0.524 | 0.064/0.065 | 0.350/0.350 | 0.7924/0.9010 |
| Total sum | 0.737/0.737 | 0.184/0.184 | | |

Note: The number after "/" is the result of variance analysis after excluding insignificant factors, *** means highly significant ($p < 0.01$), ** means significant ($0.01 < p < 0.05$), ns means not significant ($p > 0.1$).

It can be seen from the variance analysis table that the model of the air flow velocity of the seed suction hole $Y$ was extremely significant ($p < 0.01$), the thickness of the negative pressure chamber $T$ and the diameter of the transition pipe $D$ had extremely significant effects on the air flow velocity of the seed suction hole $Y$ ($p < 0.01$), and the length of the transition pipe had significant effects on the air flow velocity of the seed suction hole $Y$ ($p < 0.05$). The primary and secondary order of the three was $T > D > L$. Among the

secondary main effect items, the $D_l$ item had significant effects on $Y$ ($p < 0.05$), and the other items were not significant.

The non-significant items in the analysis of variance were excluded and the analysis of variance was conducted again. The results are shown in Table 8. The regression equation of the influence of each factor level on the air flow velocity of the seed suction hole $Y$ can be obtained as follows:

$$Y = 80.093 + 0.501T - 1.579D - 2.15l + 0.148Dl \tag{17}$$

In the analysis of variance after excluding the non-significant items, the lack of fit was 0.9010, which is not significant. It indicates that there are no other indicators affecting the test factors, the model fitting effect is good, and there is a significant quadratic relationship between the indicators and the factors. In order to obtain the interaction model between the influence of factors on the air flow velocity, the interactive response surface obtained by the software is shown in Figure 21.

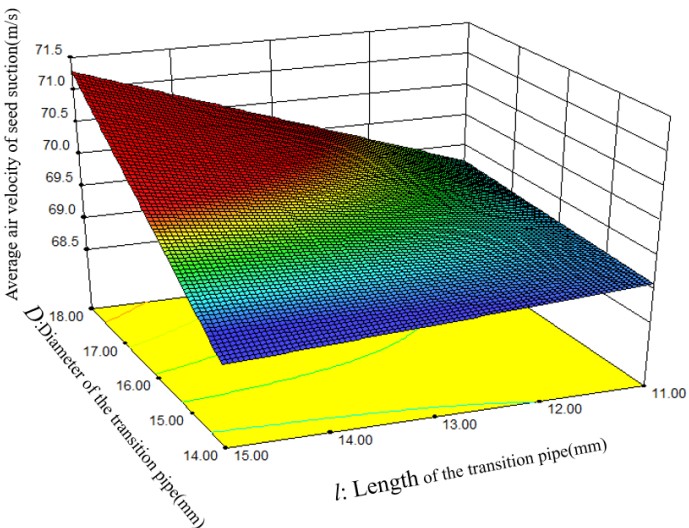

**Figure 21.** Interactive response surface of the transition pipe diameter and transition pipe length.

It can be seen from the figure that when the length of the transition pipe was large, the airflow velocity increased with the increase in the diameter of the transition pipe, and this change was obvious. However, when the length of the transition pipe was small, this change was not obvious. When the diameter of the transition pipe was large, the flow velocity increased with the increase in the length of the transition pipe. When the diameter of transition pipe was small, this change was not obvious. The interaction was obvious, consistent with the assumptions in the previous sections.

To obtain the optimal parameters for the negative pressure air chamber and air suction pipe, the Design-Expert post-processing module was used to optimize the above regression model. The objective function and constraint conditions are as follows:

$$\begin{cases} \max Y(T, D, l) \\ s.t. \begin{cases} 22\text{mm} \leq T \leq 26\text{mm} \\ 14\text{mm} \leq D \leq 18\text{mm} \\ 11\text{mm} \leq l \leq 15\text{mm} \end{cases} \end{cases} \tag{18}$$

After optimizing, the multiple groups of optimized parameter arrays were obtained. Considering the actual processing accuracy of 3D printing, the optimal parameter array was: thickness of negative pressure air chamber was 24.5 mm, diameter of transition pipe was 17.5 mm, and length of transition pipe was 14.5 mm.

### 3.3. Benchtop Experiment Verification of the Seed Drill

3.3.1. Experiment Conditions

In order to verify the rationality of the above parameter design and obtain the optimal working parameters of the seed metering device, we used 3D printing technology to complete the production of the seed drill. As shown in Figure 22, the seed drill was made from resin. In order to save costs and reduce weight, some non-stress structures were hollow to maximize the strength of the seed drill while reducing the material consumption. The traditional rubber sealing ring had poor compressibility, poor wear resistance and a high friction coefficient. After comparing a variety of materials, a polyethylene hose with a diameter of 5 mm and a thickness of 1 mm was finally selected as the main sealing material. At the same time, 0.3 mm thick silicone film was paved on the periphery. When the negative pressure began to be filled in the seed metering chamber, an effective seal was formed, and the friction coefficient between the hose and the resin was far less than that between the rubber and the resin. The assembled seed drill is shown in Figure 23, with smooth rotation and good sealing performance.

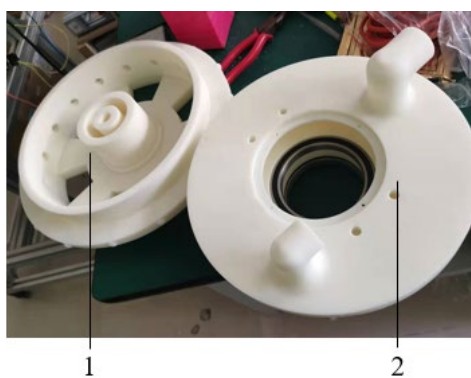

**Figure 22.** Material of the seed drill. Note: 1, Seed metering wheel; 2, Pneumatic distributor wheel.

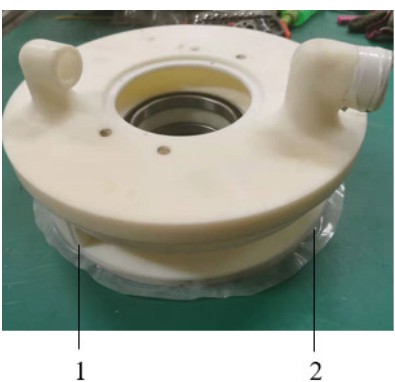

**Figure 23.** Sealing method of the seed drill. Note: 1, Polyethylene pipe; 2, Silicone film.

Selection of the appropriate fan and other components to build the experiment bench, are shown in Figure 24. The Germany Haokaide 2HB320H36 high-pressure centrifugal fan was selected as the negative pressure fan. The supporting fan controller was the Rexroth VFC3610 frequency converter, which could adjust the negative pressure value of the high-pressure fan. Then, the WS4235F-24-240-X240 fan was selected as the positive pressure fan, which could provide a maximum positive pressure of 7.5 kpa, and its air pressure could be adjusted by a controller. Lastly, a high torque DC motor was selected, with the motor rotation speed controlled by a controller.

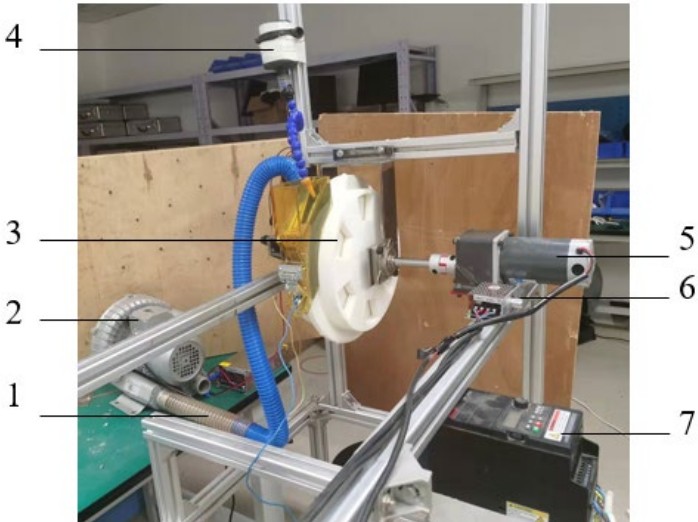

**Figure 24.** Experiment bench. Note: 1, Air suction pipe; 2, Experiment fan; 3, Sees drill; 4, Air blower; 5, DC motor; 6, DC motor controller; 7, Fan controller.

### 3.3.2. Experiment Methods and Results

We designed a three-factor and three-level an orthogonal experiment, and the experiment factors and levels are shown in Table 9.

**Table 9.** Factors and levels of bench experiment.

| Level Codes | Factors | | |
| --- | --- | --- | --- |
| | Rotational Speed of Seed Metering Wheel/$n$ (rpm) | Air Suction Negative Pressure/$P_n$ (kPa) | Seed Clearing Positive Pressure/$P_z$ (kPa) |
| 1 | 15 | 5.0 | 4.0 |
| 2 | 20 | 5.5 | 4.5 |
| 3 | 25 | 6.0 | 5.0 |

According to the agronomic requirements of wheat plot sowing, the national standard GB/T 6973-2005 *Experiment Method for Single Grain (Precision) Seeders* [33] and GB/T 10293-2013 *Technical Conditions for Single Grain (Precision) Seeders* [34], the reabsorption index $I_c$, the leakage index $I_l$ and the qualification index $I_h$ were selected as the experiment indicators. The calculation method of relevant evaluation indexes is as follows:

$$\begin{cases} I_c = \frac{n_c}{N_L} \times 100\% \\ I_l = \frac{n_l}{N_L} \times 100\% \\ I_h = 100\% - I_c - I_l \end{cases} \tag{19}$$

where $n_c$ is the number of reabsorbed, $n_l$ is the number of leakages, and $N_L$ is the theoretical number of seeds.

The determination method of reabsorption and leakage suction are shown in Figures 25 and 26. Each group of experiments was repeated three times, and the results are shown in Table 10.

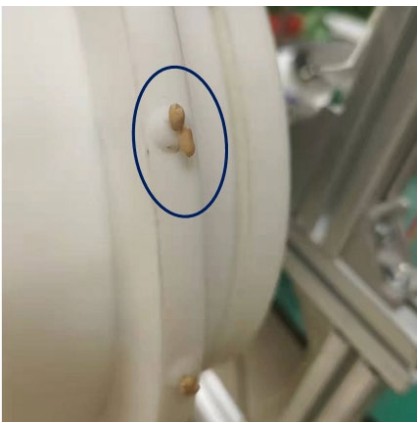

**Figure 25.** Seed reabsorption.

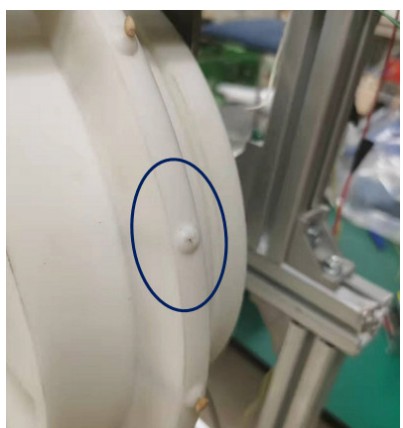

**Figure 26.** Seed leakage.

**Table 10.** Benchtop experiment results.

| Level Codes | Factors | | | Experimental Indexes | | |
|---|---|---|---|---|---|---|
| | $n$ | $P_n$ | $P_z$ | $I_c$ (%) | $I_l$ (%) | $I_h$ (%) |
| 1 | 1 | 1 | 1 | 1.62 | 6.34 | 92.04 |
| 2 | 1 | 2 | 3 | 1.58 | 6.51 | 91.91 |
| 3 | 1 | 3 | 2 | 1.51 | 6.53 | 91.96 |
| 4 | 2 | 1 | 3 | 0.82 | 6.67 | 92.41 |
| 5 | 2 | 2 | 2 | 0.91 | 6.65 | 92.34 |
| 6 | 2 | 3 | 1 | 0.94 | 6.49 | 92.47 |
| 7 | 3 | 1 | 2 | 0.79 | 7.32 | 92.09 |
| 8 | 3 | 2 | 1 | 0.88 | 7.09 | 92.03 |
| 9 | 3 | 3 | 3 | 0.85 | 7.25 | 91.90 |

The results were analyzed by range, as shown in Table 11.

**Table 11.** Range analysis of experiment results.

| Analysis Items | Reabsorption Indexes | | | Leakage Absorption Indexes | | | Qualification Indexes | | |
|---|---|---|---|---|---|---|---|---|---|
| | $n$ | $P_n$ | $P_z$ | $n$ | $P_n$ | $P_z$ | $n$ | $P_n$ | $P_z$ |
| $k_1$ | 1.57 | 1.08 | 1.15 | 6.46 | 6.78 | 6.64 | 91.97 | 92.18 | 92.18 |
| $k_2$ | 0.89 | 1.12 | 1.07 | 6.60 | 6.75 | 6.83 | 92.41 | 92.09 | 92.13 |
| $k_3$ | 0.84 | 1.10 | 1.08 | 7.22 | 6.76 | 6.81 | 92.01 | 92.11 | 92.07 |
| Optimal arrays | $n_3\ P_{n1}\ P_{z2}$ | | | $n_1\ P_{n2}\ P_{z1}$ | | | $n_1\ P_{n2}\ P_{z1}$ | | |



The basis for selecting the optimal operation parameters was the smaller the reabsorption index and the leakage suction index, the greater the qualification index, the better the operational effect of the seed drill. The range analysis showed that the rotational speed of the seed metering wheel, the negative pressure value of air suction and the positive pressure value of seed cleaning all had an impact on operational effect. Among them, the rotational speed of the metering wheel had a great impact on the reabsorption index, and the impact of air pressure on the experiment results was less than the rotational speed of the seed metering wheel. Observing the experimental process, one possible reason is that when the rotational speed of the seed metering wheel was high, the change in the instantaneous impact force of the seed suction hole on the seeds had a greater impact on above indicators than the adsorption force caused by air pressure, which made it difficult for more seeds to be adsorbed at the same time during the seed suction process. This reason also led to the influence of the speed of the seed metering wheel being greater on the reabsorption index than the leakage suction index. In order to ensure the overall effect, the selected experiment's negative pressure level was slightly higher than the theoretical negative pressure value, so the change of the leakage absorption index was not obvious.

Through the analysis of the three groups of optimal arrays obtained by range analysis, it can be seen that the best item of the factor $n$ in the reabsorption index was $n_3$, and the best item of the leakage suction index and the qualification index was $n_1$. The influence of the rotational speed of the seed metering wheel on the operational effect of the seed drill was mainly reflected in the reabsorption index. $n_3$ was 0.73% better than $n_1$ and 0.05% better than $n_2$, with little difference. The leakage absorption index $n_1$ was 0.76% better than $n_3$ and 0.14% better than $n_2$. The qualification index $n_1$ was 0.96% better than $n_3$ and 0.56% better than $n_2$. It was found that among the three indicators, the difference between the $n_2$ item and the optimal item was small, and the reabsorption index was very close to the optimal item. Therefore, the $n_2$ term was selected as the best when considering the three indicators comprehensively.

The optimal factor $P_n$ was $P_{n1}$ in the leakage absorption index, and $P_{n2}$ in the reabsorption index and qualification index was the best. $P_{n1}$ was better than $P_{n2}$ by 0.04% in the reabsorption index, and $P_{n2}$ was better than $P_{n1}$ by 0.03% in the leakage absorption index. Above all, $P_{n1}$ was the best.

The best factor $P_z$ was $P_{z2}$ in the leakage absorption index, and $P_{z1}$ in the reabsorption index and qualification index was the best. $P_{z2}$ was superior to $P_{z1}$ by 0.08% in the reabsorption index. Although $P_{z1}$ was superior to $P_{z2}$ by 0.19% in the reabsorption index, but it was only superior to $P_{z2}$ by 0.05% in the qualification index. In combination with the principle of "better missed sowing and no reseeding", $P_{z2}$ was the best. Therefore, the final optimal operation parameter group was $n_2$ $P_{n1}$ $P_{z2}$, with the rotation speed $n$ of the seed metering wheel at 20 rpm, the negative pressure $P_n$ of seed suction at $-5.0$ kPa, and the positive pressure $P_z$ of seed cleaning at 4.5 kpa. At this time, the reabsorption index was 0.82%, the leakage absorption index was 6.67%, and the qualification index was 92.41%, meeting the design requirements.

## 4. Conclusions

(1) In this paper, a single seed drill with air suction wheel holes based on the agronomic requirements of precision sowing in wheat plots was designed. This paper analyzed its working principle and process and determined its key structural parameters. Additionally, the diameter of the seed metering wheel was designed to be 180 mm, with 16 seed suction holes.

(2) Measuring the physical parameters of the wheat seeds used in the experiment and using the discrete element simulation software EDEM to analyze the seed disturbance effects of different parameter designs of the seed metering wheel. Then, through the three-factor and three-level orthogonal experiment, the final design scheme was determined as follows: a thickness of the seed suction convex ring of 16 mm, height of the seed suction mouth of 4.5 mm, and correlation coefficient $k_i$ of 0.75, with the diameter of the seed suction cam at 12 mm.

(3) The air force distribution of the seed drill was analyzed. The phase angle of the negative pressure chamber was determined to be 280 °, the phase angle of the positive pressure chamber was 22 °, and the phase angle of the non-pressure interval was 20 °.

(4) Analyzing the force of seed suction at the seed suction hole, the theoretical seed suction negative pressure value was determined to be −5 kpa. The ANSYS Workbench simulation platform was used to carry out hydrodynamic simulation analysis of the negative pressure basin. Three-factor and three-level quadratic rotation orthogonal experiment was conducted to study the influence of the seed suction pipe and the negative pressure chamber parameters on the air flow velocity of the seed suction hole. Then, through Design-Expert data analysis software was used to find the influence of the thickness of the negative pressure chamber, the diameter of the transition pipe and the length of the transition pipe on the airflow velocity in the seed suction hole. The final scheme was selected as follows: a thickness of the negative pressure air chamber of 24.5 mm, diameter of the transition pipe of 17.5 mm, and a length of the transition pipe of 14.5 mm.

(5) The seed drill was made by 3D printing technology, and the suitable sealing material was selected for benchtop experiment verification. A Three-factor and three-level orthogonal experiment was designed to analyze the influence of different parameters on the seed suction effect of the seed drill. The final working parameters were determined as follows: a rotational speed of the seed metering wheel of 20 rmp, negative pressure of seed suction of −5 kpa, and positive pressure of seed cleaning of 4.5 kpa. At this time, the reabsorption index was 0.82%, the leakage suction index was 6.67%, and the qualified index was 92.41%, which met the design requirements.

**Author Contributions:** Conceptualization, methodology, software and data curation, X.M.; supervision and project administration, Q.W. and D.X.; formal analysis, Y.Z.; validation and visualization, X.M., Q.G., G.C., X.C. and L.W.; writing—original draft preparation, X.M. and Y.Z.; writing—review and editing, Y.Z. All authors have read and agreed to the published version of the manuscript.

**Funding:** This research received no external funding.

**Institutional Review Board Statement:** Not applicable.

**Data Availability Statement:** Not applicable.

**Conflicts of Interest:** The authors declare no conflict of interest.

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
