# Peer review of "Design of an Air Suction Wheel-Hole Single Seed Drill for a Wheat Plot Dibbler"

_agriculture, doi:10.3390/agriculture12101735_

Round 1
Reviewer 1 Report
Although the techniques used in this paper, including the methods of measurement and analysis are very well-known and have been reported many times previously, how the information or experimental data was analyzed and discussed appropriately rather than the methods or techniques used is perhaps most important criteria to evaluate this paper. In this regard, the information presented here is not acceptable to publish in the journal. Nevertheless, there are some minor and major concerns to be solved or answered. The following comments are shown.
1. I suggest to revise the full manuscript with a native English speaker before consider for publication.
2. Abstract can be more succinctly written. Abstract should be expanded sentences related to the results. The results of the study should be given as numerical percentages.
3.The paper should be also supported by a literature search including relevant and recent papers. The following articles related to optimization and EDEM analyses may be cited.
4.The last question is that have the comparison between the designed seed drill device and other devices been done? How about the highest efficiency of the device.
5. The sentence ”In order to realize single-seed dibbling, most plot breeding experiments in China still adopt artificial seeding [4-5]”, it shoule be reviesd.
6.The sentence“In the above formula,”should be revised.
7. In the section of conclusion, there is a format error.
8.In the same way, including a photograph is recommended that gives certainty that the measurement systems was used.
9. Please re-write the Statistical analysis to explain the models, factors, treatments that used.
10. Would you explicitly specify the novelty of your work? What progress against the most recent similar studies was made?
11 Fig.21 test method is not clear enough.
10 The quality of the figures is not good and should be improved. In all figures, Increase font size for better readability.
Author Response
Response to Reviewer 1 Comments
Point 1: I suggest to revise the full manuscript with a native English speaker before consider for publication.
Response 1: Dear reviewer, I'm sorry that we don't have any foreigners we are familiar with. But the content of the manuscript has been rectified as much as possible.Please see the attachment.
Point 2: Abstract can be more succinctly written. Abstract should be expanded sentences related to the results. The results of the study should be given as numerical percentages.
Response 2: In the new manuscript, the summary section has been changed.
Point 3: The paper should be also supported by a literature search including relevant and recent papers. The following articles related to optimization and EDEM analyses may be cited.
Response 3: The content of EDEM simulation design optimization in this paper was proposed by us, so no relevant literature was cited.
Point 4: The last question is that have the comparison between the designed seed drill device and other devices been done? How about the highest efficiency of the device.
Response 4: The air suction wheel hole single seed metering device designed by us is only applicable to single seed sowing of wheat, and has a high degree of uniqueness. Therefore, there is no similar metering device at present, and it is hard to compare the same type. The qualification index in the bench experiment of this equipment, namely the sowing efficiency, is 92.41%, which meets the design requirements.
Point 5: The sentence ”In order to realize single-seed dibbling, most plot breeding experiments in China still adopt artificial seeding [4-5]”, it shoule be reviesd.
Response 5: Thank you for criticism and correction. We have made relevant modifications in the new manuscript.
Point 6: The sentence“In the above formula,”should be revised.
Response 6: Thank you for criticism and correction. We have made relevant modifications in the new manuscript.
Point 7: In the section of conclusion, there is a format error.
Response 7: Thank you for criticism and correction. We have made relevant modifications in the new manuscript.
Point 8: In the same way, including a photograph is recommended that gives certainty that the measurement systems was used.
Response 8: I'm sorry that no photos were taken during that experiment, and only the relevant results were recorded.
Point 9: Please re-write the Statistical analysis to explain the models, factors, treatments that used.
Response 9: In the new manuscript, the contents of relevant parts have been modified.
Point 10: Would you explicitly specify the novelty of your work? What progress against the most recent similar studies was made?
Response 10: The greatest feature of the seed metering device designed by us is that it is an innovative and mature pneumatic seed metering device with simple structure and high working stability on the premise of meeting the demand of single seed on-demand sowing in wheat plot. Compared with the recent research, the biggest progress is to solve the problem that the mechanical wheat seed metering device has poor seed filling performance and poor seed metering stability.
Point 11: Fig.21 test method is not clear enough.
Response 11: In the new manuscript, the contents of relevant parts have been modified.
Point 12: The quality of the figures is not good and should be improved. In all figures, Increase font size for better readability.
Response 12: To solve this problem, we enlarged some pictures that are difficult to see in the article to improve the clarity of expression.

Reviewer 2 Report
The authors wrote the paper in a long-winded way. Its character is rather technical than scientific. Some parts of abstract and description of the machine are difficult to understand..The analysis of data in Fig.14 is difficult. The figures have a wrong format.
Author Response
Dear reviewer, thank you for your criticism and correction. We have made relevant modifications to the content of the machine description. Figure 14 has been enlarged to improve readability. And the language sturcture of the manuscript has been rectified as much as possible. Please see the attachment.

Reviewer 3 Report
This paper designed a special air suction wheel hole single seed drill for remote control self-propelled single seed dibbler in wheat plot. The manuscript is well organized with enormous data and suitable for publication in the Agriculture journal after improving the study gap and novelty of the work in the introduction section.
Author Response
Dear reviewer, thank you for your reviewing. We have recitified the language sturcture of the manuscript as much as possible. Please see the attachment.

Round 2
Reviewer 1 Report
The author has revised all my questions, but the English expression needs to be further revised.
Author Response
Dear reviewer
Thank you for your guidance and suggestions. We have improved the language expression of the research results and conclusions in this paper.
We wish you a happy life and smooth work! Kind regards, Yinggang Zhou
Reviewer 2 Report
I have not any special remarks to improoved paper.
Author Response
Dear Reviewer,
Thank you for your guidance and suggestions. We wish you a happy life and smooth work!
Kind regards, Yinggang Zhou